# Research on the Effect of Spindle Speed on the Softening and Hardening Characteristics of the Axial Operating Stiffness of Machine Tool Spindle

**Jiandong Li** [1,2,*] **, Qiang Wang** [1] **, Xurui Sun** [3] **, Jue Qu** [2] **, Ang Qiu** [2] **, Wei Kang** [4] **and Shuaijun Ma** [4]

1. Equipment Management and Unmanned Aerial Vehicle Engineering School, Air Force Engineering University, Xi'an 710051, China; caption_wang@21cn.com
2. Air and Missile Defend School, Air Force Engineering University, Xi'an 710051, China; qujue402@sina.com (J.Q.); qiuanng@163.com (A.Q.)
3. Baqiao District Economic Information Research Center, Baqiao District Development and Reform Commission, Xi'an 710051, China; sunxurui@126.com
4. Key Laboratory of Education Ministry for Modern Design and Rotor-Bearing System, Xi'an Jiaotong University, Xi'an 710049, China; kang_wei@stu.xjtu.edu.cn (W.K.); msj821@stu.xjtu.edu.cn (S.M.)
* Correspondence: zuiwojinxiao@stu.xjtu.edu.cn

**Abstract:** Spindle stiffness is one of the most critical indicators for evaluating and measuring the service performance of spindles. The traditional static stiffness indexes only involve static analysis and rarely focus on the study of spindle-carrying capacity under operating conditions. In this paper, the explicit solution approach is used to develop a mechanical model of the spindle's axial operating stiffness. This model was then used to explore the influence of rotational speed on the softening and hardening features of the spindle axial operating stiffness, and experimental verification was carried out. According to studies, the speed of a fixed-position preload spindle can lead its operating stiffness to exhibit a "stiffness-hardening" feature. However, when the axial displacement of the spindle is small, the operating stiffness curve of the spindle displays a noticeable "fluctuation" phenomenon for low-speed spindles. Furthermore, the speed-induced preload has a significant impact on the test results when testing spindle axial operating stiffness.

**Keywords:** axial operating stiffness; stiffness hardening; stiffness softening; machine tool





## 1. Introduction

Cutting forces are the most common external loads applied to machine tool spindles and are widely regarded as the best performance estimator for machining operations [1–3]. Spindle stiffness of the machine tool, which reflects its capacity to resist deformation when subjected to external loads, is one of the most essential indicators for evaluating the service performance of a spindle [4–6]. Low spindle stiffness causes chatter [1], unwanted back cutting, and excessive cutter tilt, all of which affect the cutting surface quality [7–9] and the machining accuracy [10,11], as well as causing the rolling elements of the spindle bearing to slip, aggravating spindle component wear [12] and weakening machine tool reliability [13–15]. As a result, high stiffness becomes one of the most important aims in precision machine tool spindle design [16].

Spindle static stiffness is a regularly used metric for measuring and evaluating spindle stiffness under constant or slowly variable quasi-static loads [17]. Static and dynamic analysis are two approaches to study spindle static stiffness [18]. The static analysis of spindle static stiffness has been investigated previously and is considered to be rather advanced. D. Olvera [18] proposed a static stiffness measurement method along the turning center kinematic chain, and analyzed the tool tip radial stiffness of turn-milling centers. With the help of a loaded double-ball bar and linear variable differential transformer, Laspas [19] proposed a new method to measure and identify the full translational stiffness

matrices of the five-axis machining center by using the quasi-static circular trajectory, and realized the accurate identification of the quasi-static stiffness of the five-axis machine tool. As the accuracy of spindle machining improves, the spindle's service speed must also increase. When the spindle is rotating, however, the state characteristics are very different from when it is halted. Based on the proposed thermo-mechanical model of the spindle system, Li [20] discovered that while the spindle is operating at high speed, the clearance of its components is much different from that at standstill, and this has a considerable impact on spindle preload and component temperature. Subsequently, Li [21] then evaluated the transient preload of a fixed-position preload spindle in real-time and discovered that at 8000 rpm, the preload rose from 483 N at standstill to 720 N, while the bearing temperature increased from 24 °C to 36.5 °C. Large variations in the machine spindle's characteristics during operation are sure to impair the spindle stiffness performance, which, in turn, affects the machine tool's quality and efficiency. As a result, the dynamic analysis of spindle static stiffness (in this paper, referred to as operating stiffness to distinguish it from static and dynamic stiffness) is gaining attention.

In order to better investigate the effect of dynamic effects generated by spindle operation on its static stiffness, A. Matsubara [22] designed a magnetic loading device to measure the radial operating stiffness of the spindle and found that the velocity and thermal effects can lead to significant softening and hardening characteristics in the spindle radial stiffness. Wang [10] proposed a three-step identification algorithm for spindle radial stiffness based on stiffness theory modeling, which solves the difficult problem of the measured value being often coupled in the spindle-tool stiffness during spindle radial operating stiffness testing, and improves the accuracy of spindle radial operating stiffness testing. For drilling machines, vertical spindle surface grinders, and other axially loaded machine tools, the axial operating stiffness of the spindle system should be of increased importance throughout its entire life cycle. Tsuneyoshi [23] discovered that the spindle axial load–axial displacement curve showed a non-linear relationship in his investigation into spindle preload testing methodologies. Li [4] further explored the non-linear relationship of the spindle axial load–displacement curve and found that the machine tool spindle static stiffness exhibits hardening and softening characteristics depending on the preload. However, the speed-induced centrifugal effect generates a non-linear change in bearing stiffness, which will inevitably lead to a change in spindle stiffness, as the bearing stiffness determines the overall stiffness of the spindle bearing [24,25]. It is a pity that the influence of spindle speed on the softening and hardening characteristics of the axial operating stiffness of machine tool spindles has not been discovered in any of the preceding studies.

In this paper, the spindle axial operating stiffness of a widely configured fixed-position preload spindle for precision machine tools was investigated. The explicit solution approach is used to develop a mechanical model of the spindle's axial operating stiffness. This model was then used to explore the influence of rotational speed on the softening and hardening features of the spindle axial operating stiffness. A spindle operating stiffness test bench was also created to evaluate the model's validity and accuracy, with the impact of speed-induced preload on spindle operating stiffness being studied in particular.

## 2. Axial Operating Stiffness Model of Spindle

In this section, an explicit solution approach is used to build an angular contact ball-bearing mechanics model and analyze the relationship between speed, preload, and contact angle. Following that, an analytical model of the fixed-position preload spindle operating stiffness is constructed based on the spindle preload principle and the aforesaid bearing mechanics model. The influence of speed on the axial operating stiffness of the machine tool spindle can be investigated by adjusting the speed parameter in the spindle stiffness model.

### 2.1. Bearing Mechanics Model

Different service conditions, such as external load and speed, vary the bearing contact angle. In general, when a bearing is assembled, a certain preload is given to minimize

bearing clearance to achieve the desired bearing stiffness, accuracy, and other characteristics [26,27]. The bearing contact angle will change from the initial contact angle $\alpha_f$ to $\alpha_p$, and the outer raceway groove curvature center, ball center, and inner raceway groove curvature center will all be co-linear at this point (as shown in Figure 1). When the bearing rotates, the centrifugal force and gyroscopic moment effect generated by the speed act on the ball, causing the outer raceway groove curvature center, ball center, and inner raceway groove curvature center to lose co-linearity, and the ball-inner contact angle $\alpha_i$ and the ball-outer contact angle $\alpha_o$ to no longer be equal.

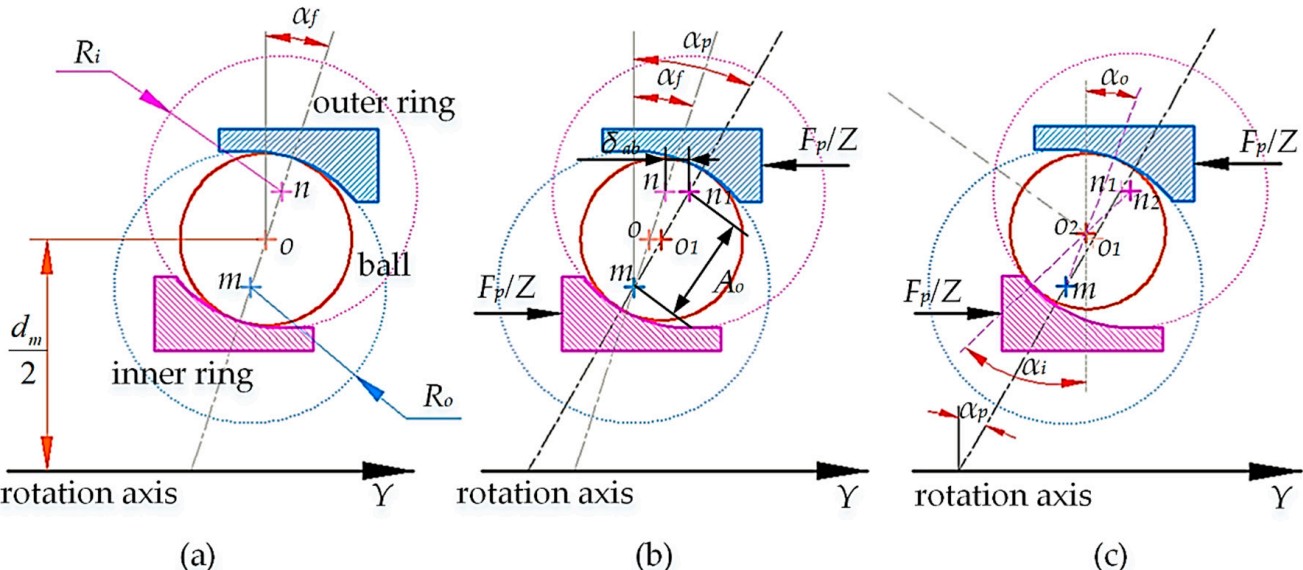

**Figure 1.** Bearing contact angle variation diagram: (**a**) free state; (**b**) withstand preload; (**c**) rotation status.

According to Ref. [4], the contact angle of the bearing with preload applied can be calculated as:

$$\alpha_p = \arcsin\left(\frac{A_o \sin \alpha_f + \delta_{ab}}{\sqrt{A_o^2 \cos^2 \alpha_f + \left(A_o \sin \alpha_f + \delta_{ab}\right)^2}}\right) \tag{1}$$

where $\alpha_p$ means the bearing contact angle with preload applied; $\alpha_f$ indicates the initial contact angle of the bearing, $A_o$ is the distance between raceway groove curvature centers; and $\delta_{ab}$ means the axial displacement of the bearing.

When the bearing rotates, the ball is subjected to centrifugal force and gyroscopic moment, as shown in Figure 2. The centrifugal force can be split into two components: the component force $F_u$ in the normal direction and the component force $F_w$ parallel to the tangential direction of the contact point. The bearing load distribution is affected differently by the two components of centrifugal force. The component force $F_u$ increases the normal load in the bearing outer ring on the ball, whereas the component force $F_w$ compresses the ball farther against the bearing inner and outer rings. The ball center travels along the force's direction when component force $F_w$ is applied. At this time, the ball-inner and the ball-outer contact angles are no longer equal, satisfying Equation (2).

$$\frac{1}{\tan \alpha_o} - \frac{1}{\tan \alpha_i} = \frac{7}{5} \frac{Z F_c}{F_p} \tag{2}$$

where $Z$ indicates the number of balls in the bearing; $F_p$ means preload applied to bearing; and $F_c$ refers to centrifugal force, which can be calculated from Equation (3).

$$F_c = \frac{d_m}{2} m_b \omega_c^2 \tag{3}$$

where $d_m$ denotes diameter of bearing pitch circle; $m_b$ refers to ball mass; and $\omega_c$ is bearing cage speed.

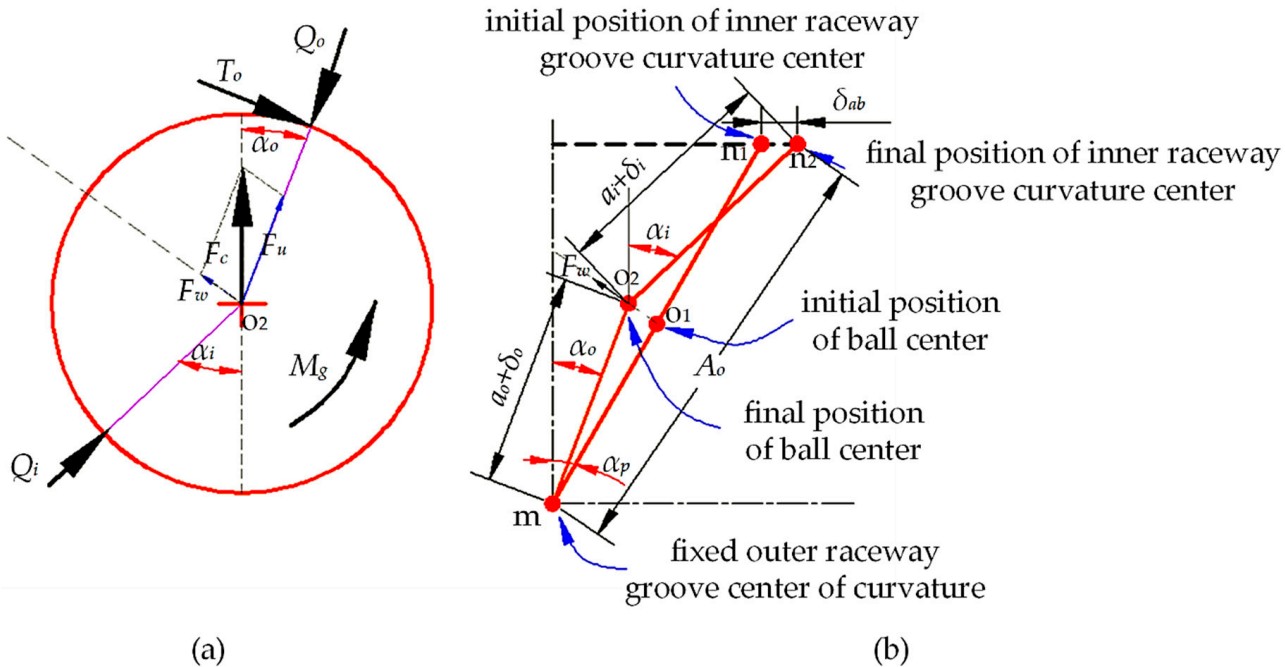

**Figure 2.** Parameter variation and load distribution on the ball after bearing rotation: (**a**) bearing ball load distribution; (**b**) variation in bearing groove curvature center position.

In order to construct a system of equations for calculating the contact angle $\alpha_o$ and $\alpha_i$, another set of functional relationships is required in addition to Equation (2). According to the relationship between the curvature position of each bearing groove shown in Figure 2b, the contact angles $\alpha_o$ and $\alpha_i$ can also be derived from Equation (4).

$$(a_o + \delta_o) \cos \alpha_o + (\alpha_i + \delta_i) \cos \alpha_i = (a_o + a_i) \cos \alpha_f \tag{4}$$

where $a$ indicates the distance between the raceway groove curvature center and the ball center and $\delta$ indicates the ball-race deformation. The subscripts $o$ and $i$, respectively, concern outer ring and inner ring.

According to the explicit solution algorithm of Ref. [28], the ball-inner contact angle $\alpha_i$ and the ball-outer contact angle $\alpha_o$ can be obtained by combining the system of Equations (2) and (4).

The bearing axial displacement $\delta_{ab}$ under the preload force $F_p$ can be obtained by substituting the results of the contact angles $\alpha_o$ and $\alpha_i$ into Equation (5).

$$\delta_{ab} = (a_o + \delta_o) \sin \alpha_o + (a_i + \delta_i) \sin \alpha_i - (a_o + a_i) \sin \alpha_f \tag{5}$$

*2.2. Spindle Axial Operating Stiffness Model*

With fixed-position preload, the combined bearing's axial relative position remains constant throughout use. Figure 3 depicts a schematic representation of the fixed-position preload spindle construction. When the spindle bearings are mounted back-to-back, the variation in width between the inner and outer spacers can modify the preload.

According to the analysis in Section 2.1, the relationship between the axial load $F_{ab}$ and its corresponding axial displacement $\delta_{ab}$ can be obtained. Here, $F_{ab}$ and $\delta_{ab}$ satisfy the following relationship for convenience:

$$F_{ab} = f(\delta_{ab}, n) \tag{6}$$

where $f(\cdot)$ represents the non-linear mapping relationship and $n$ refers to the spindle bearing speed.

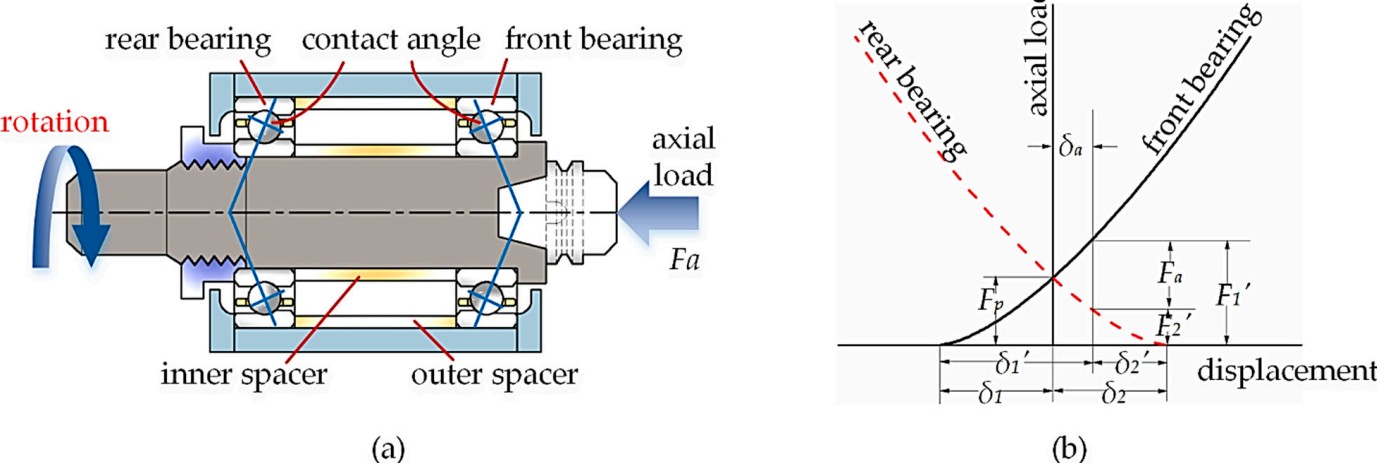

**Figure 3.** One side of the spindle is not completely unloaded during the pressure phase: (**a**) fixed position preload spindle construction; (**b**) spindle load distribution.

As shown in Figure 3b, the initial preload of the spindle is assumed to be $F_p$, and the load distribution of its front and rear bearings satisfies the following relationship:

$$F_p = f(\delta_1, n) = f(\delta_2, n) \tag{7}$$

where $\delta_1$ donates the pre-deflection of the front bearing and $\delta_2$ means pre-deflection of the rear bearing.

The spindle system is gradually subjected to axial pressure, which increases the stress on the front bearing of the spindle while lowering the force on the rear bearing. When the axial displacement of the spindle $\delta_a$ is less than the pre-deflection of the rear bearing $\delta_2$, the force relationship of the spindle is given as:

$$f(\delta_1 + \delta_a, n) = F_a + f(\delta_2 - \delta_a, n) \tag{8}$$

Continue applying axial pressure until the spindle axial displacement $\delta_a$ equals or surpasses the rear bearing pre-deflection $\delta_2$ (as shown in Figure 4), the rear bearing is completely unloaded, and the axial load is borne entirely by the spindle's front bearing. The force relationship for the spindle can now be expressed as:

$$f(\delta_1 + \delta_a, n) = F_a \tag{9}$$

The analytical process remains the same when the spindle is subjected to axial tension. The spindle system is gradually supplied axial tension, and as the force on the rear bearing grows, the force on the front bearing of the spindle diminishes. Before the front bearing of the spindle is entirely unloaded (i.e., the axial displacement of the spindle $\delta_a$ is less than the pre-deflection of the front bearing $\delta_1$), so the force relationship of the spindle is as follows:

$$F_a + f(\delta_1 - \delta_a, n) = f(\delta_2 + \delta_a, n) \tag{10}$$

Continue to apply axial tension until the axial displacement $\delta_a$ of the spindle is greater than or equal to the pre-deflection of the front bearing $\delta_1$, at which point the front bearing of the spindle is completely unloaded and the axial load is completely borne by the rear bearing of the spindle. The force balance equation of the spindle can be given as:

$$F_a = f(\delta_2 + \delta_a, n) \tag{11}$$

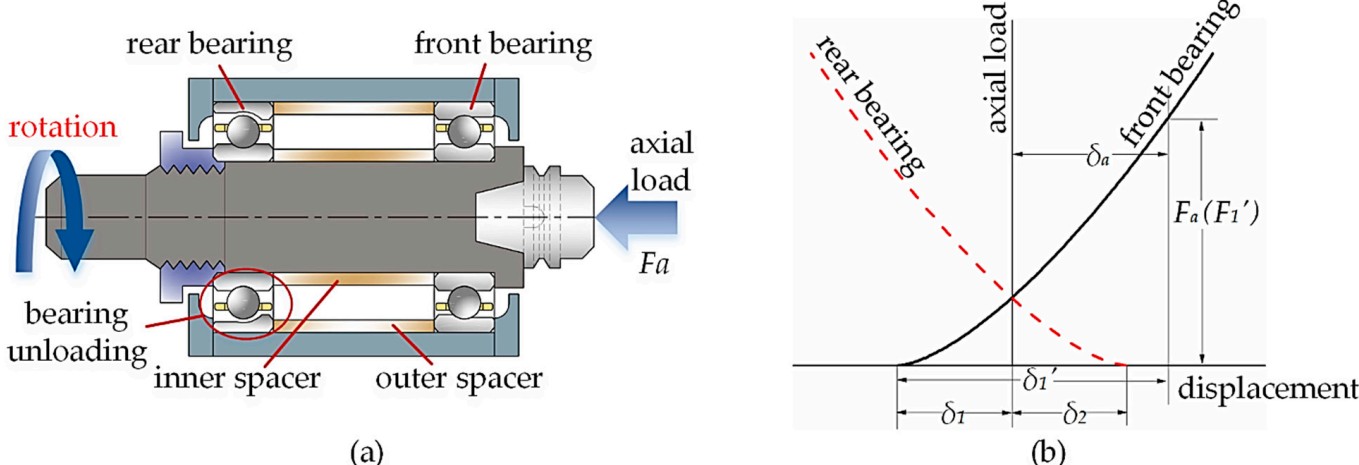

**Figure 4.** One side of the spindle is completely unloaded during the pressure phase: (**a**) fixed-position preload spindle construction; (**b**) spindle load distribution.

By solving the preceding Equations (7)–(11) together, the relationship between axial load applied to spindle $F_a$ and its associated axial displacement $\delta_a$ can be obtained. Subsequently, the spindle axial operating stiffness $K_a$ can be calculated, as given in Equation (12).

$$K_a = \frac{\mathrm{d}F_a}{\mathrm{d}\delta_a} \tag{12}$$

## 3. Effect of Spindle Speed on Spindle Axial Operating Stiffness

According to Ref. [4], the preload has a softening and hardening influence on the axial static stiffness of the spindle. Therefore, this paper explores the effect of speed on the axial operating stiffness of the fixed-position preload spindle by adjusting the spindle bearing speed parameters in the model under working circumstances with varying initial preload, with simulation results given in Figures 5 and 6. Among them, the bearing model utilized in the theoretical simulation is NSK®7014CTYNSULP4, the dimensional parameters of which are provided in Table 1.

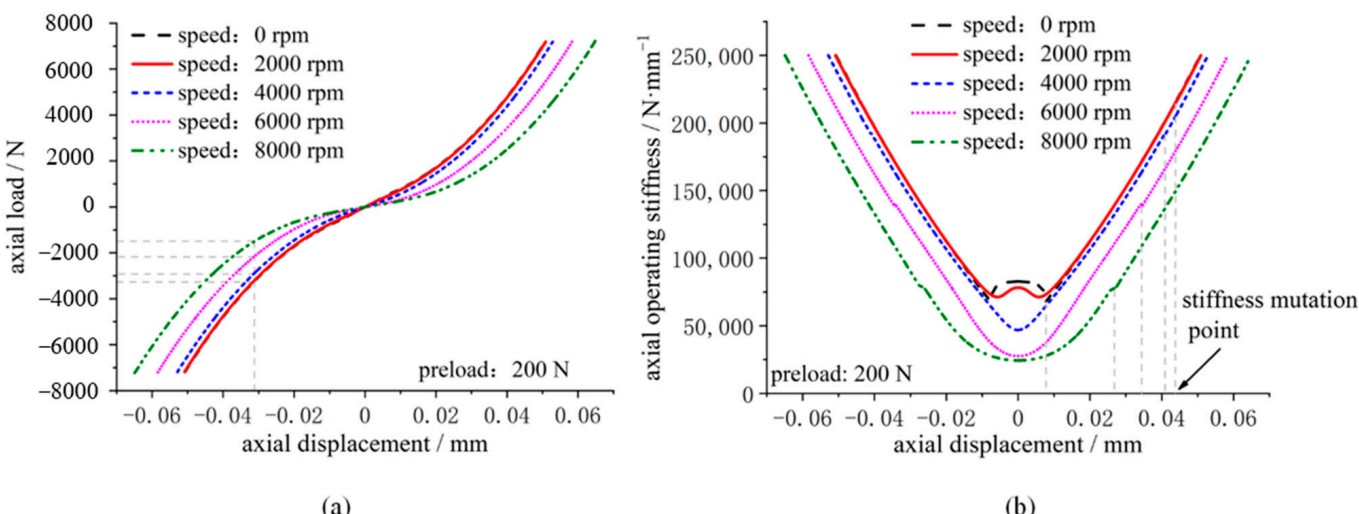

**Figure 5.** Effect of speed on axial operating stiffness of fixed-position preload spindle (preload: 200 N): (**a**) axial load–displacement curve; (**b**) axial operating stiffness curve.

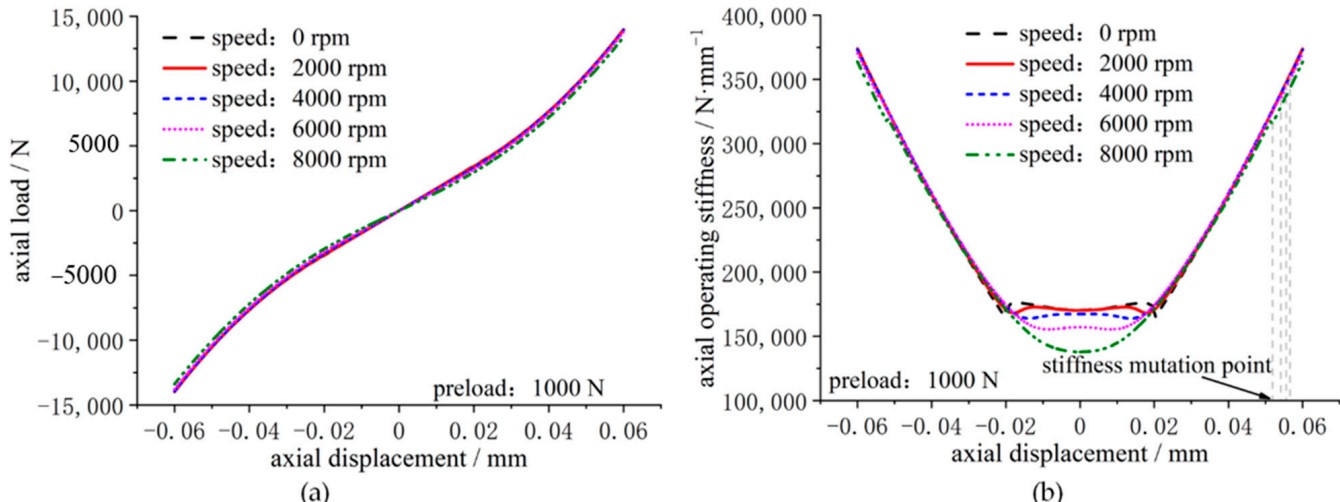

**Figure 6.** Effect of speed on axial operating stiffness of fixed-position preload spindle (preload: 1000 N): (**a**) axial load–displacement curve; (**b**) axial operating stiffness curve.

**Table 1.** The bearing parameters (NSK®7014CTYNSULP4).

| Parameters | Values |
|---|---|
| Number of balls | 20 |
| Ball diameter | 11.906 mm |
| Bearing pitch diameter | 90 mm |
| Inner raceway groove curvature radius | 6.19112 mm |
| Outer raceway groove curvature radius | 6.19112 mm |
| Initial contact angle of bearing | 15° |

Figure 5 shows that for the fixed-position preload spindle with a smaller initial preload, the rotational speed has a greater influence on the load–displacement relationship and axial operating stiffness (200 N). As seen in Figure 5, as the speed increases, the load–displacement curve of the spindle becomes "smoother," meaning that the overall axial stiffness of the spindle reduces. In comparison to Figure 6, the spindle speed has less effect on the load–displacement relationship and axial operating stiffness when the spindle's initial preload is greater (1000 N). Because the bearing stiffness "softening" effect caused by rotational speed is more noticeable when the spindle preload is low, and the fixed-position preload spindle stiffness is a parallel relationship between the spindle front and rear bearing stiffness, the higher the spindle speed is, the lower the overall axial stiffness is. As shown in Figure 7, when the spindle preload is higher, the "softening" effect of bearing stiffness due to rotational speed is lessened, as is its effect on overall spindle stiffness.

Furthermore, as shown in Figure 5b, when the spindle is rotated, the axial displacement at the stiffness mutation point is greater than the axial displacement at zero speed. In the meantime, when the rotational speed increases, the axial displacement of the spindle's axial operating stiffness mutation point decreases.

The following are the reasons behind this: When the initial preload of the spindle remains constant, the initial pre-deflection of the bearing decreases as the spindle speed increases, but is greater than the pre-deflection at zero speed, which is the macroscopic manifestation of the bearing "stiffness softening" effect caused by the spindle speed (as shown in Figure 8). The spindle axial displacement must entirely balance the spindle's initial pre-deflection in order for the abrupt change in spindle axial operational stiffness to occur. As a result, the axial displacement at the stiffness mutation point is larger than the axial displacement at zero speed when the spindle is rotated. At the same time, when the speed of the spindle rises, the axial displacement at the stiffness mutation point decreases.

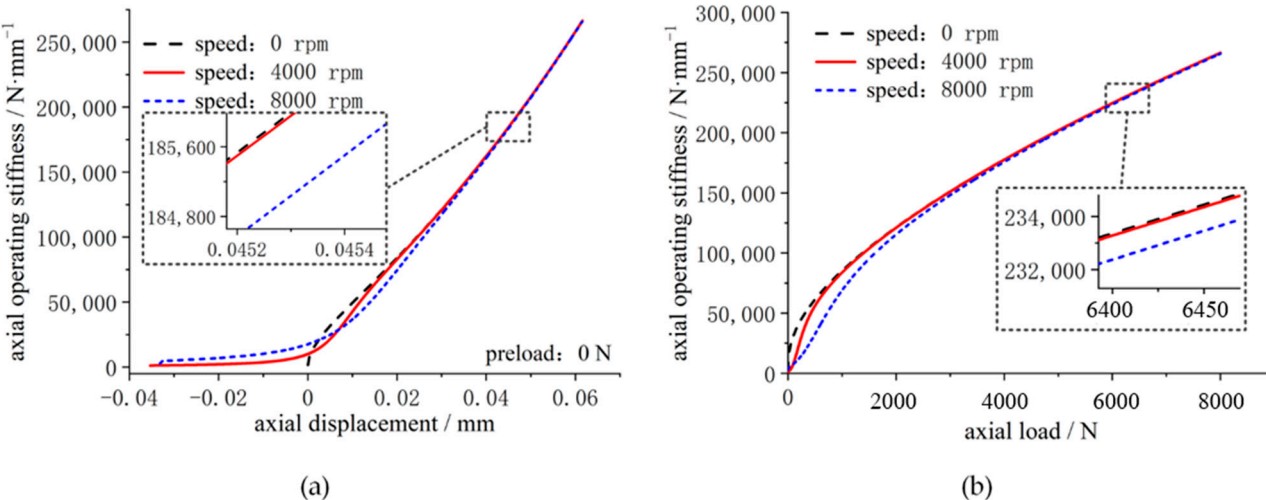

**Figure 7.** Single bearing axial operating stiffness at different speeds (bearing type: 7014C): (**a**) axial operating stiffness–axial displacement curve; (**b**) axial operating stiffness–axial load curve.

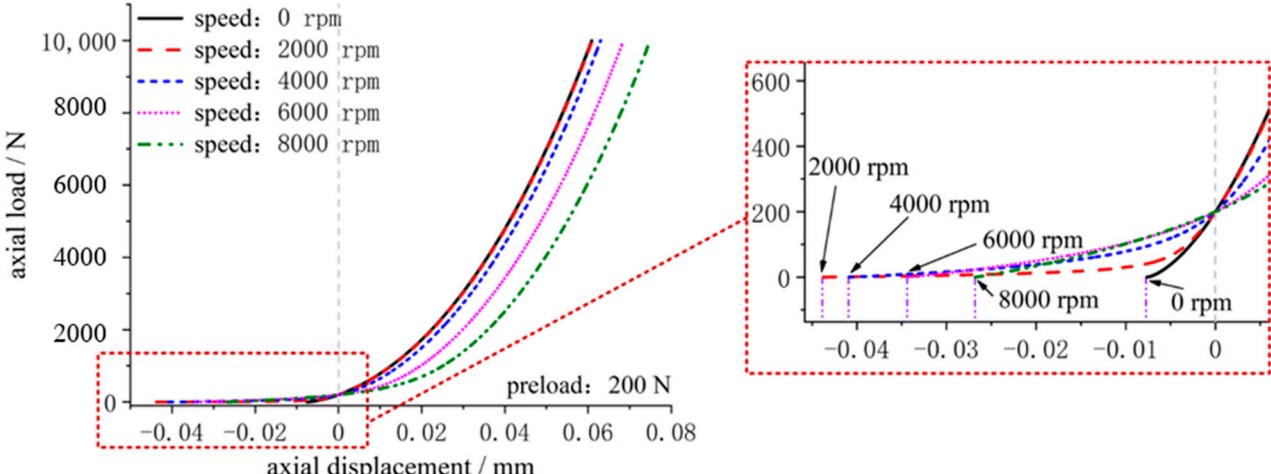

**Figure 8.** Single bearing load–displacement curve at different speeds (preload: 200 N).

It can also be observed in Figures 5 and 6 that when the spindle speed increases, the "sag" [1] in the axial stiffness curve of the spindle becomes more prominent (i.e., the spindle axial stiffness increases with the gradual increase in axial displacement before the abrupt change in spindle stiffness). When the spindle is rotating slowly, the axial stiffness of the spindle fluctuates (not exactly monotonically increasing or decreasing) in the phase with small displacement increments (e.g., −0.02 mm to 0.02 mm in Figure 6b). However, as the spindle speeds up, the stiffness fluctuation is gradually suppressed. When the spindle is operating at high speeds, the spindle axial stiffness displays a clear "sag" phenomenon, which shows the "stiffness hardening" characteristic.

This is because while the spindle is under axial load, one side bearing of the spindle progressively compresses while the other gradually unloads. Given that the fixed-position preload spindle stiffness is equal to the parallel connection of the front and rear bearing stiffness (superposition relationship) [4], the different change rates of single bearing stiffness with speed will result in different softening and hardening characteristics presented by the spindle axial operating stiffness.

When the spindle is at a lower speed, the local maximum and local minimum points in the bearing stiffness change rate curve are relatively obvious (as shown in the curve of the speed of 4000 rpm in Figure 9). The spindle pre-compression displacement value is typically positive because the spindle bearing in service requires a sufficient preload to keep

the inner and outer rings and balls in contact. When the spindle is loaded, the relationship between the spindle pre-compression and the local maximum and local minimum values of the change rate in bearing stiffness differs, resulting in the increase rate in bearing stiffness on one side that differs from the decrease rate in bearing stiffness on the other. Before the spindle bearing is completely unloaded, the spindle stiffness depends on the superposition effect of the increase rate of the one-side bearing stiffness and the decrease rate of the other-side bearing stiffness. At this time, there is a certain fluctuation of the spindle axial operating stiffness (i.e., exhibiting the stiffness softening and hardening effect). When the spindle bearing is completely unloaded, the spindle stiffness is transformed into the single-side bearing stiffness, and the spindle axial stiffness increases with the gradual increase in axial displacement (i.e., the spindle shows stiffness hardening characteristics).

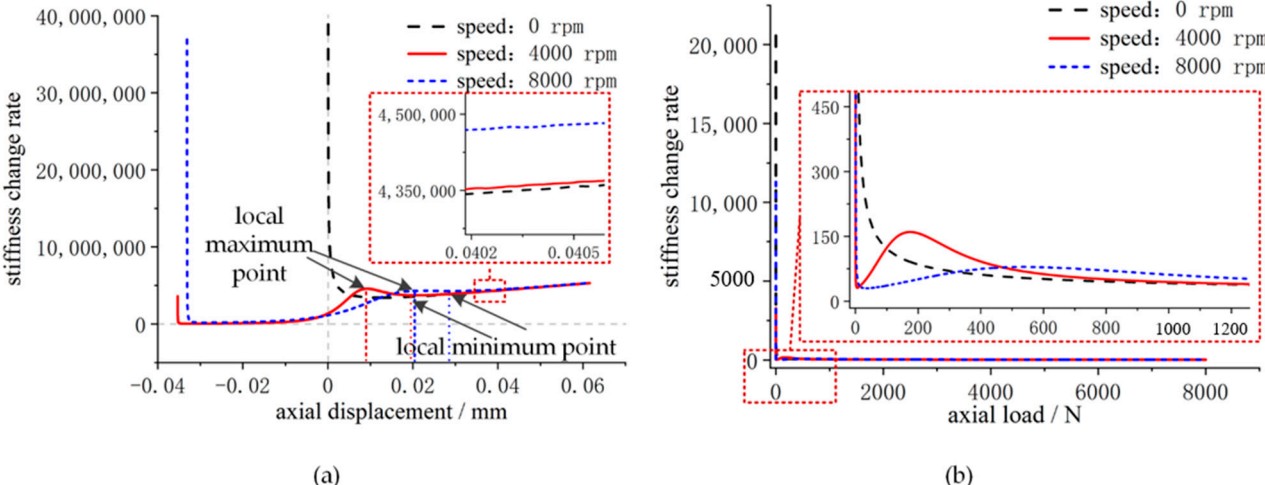

**Figure 9.** Single bearing stiffness change rate (corresponds to Figure 7): (**a**) relationship between stiffness change rate and axial displacement; (**b**) relationship between stiffness change rate and axial load.

When the spindle is at a higher speed, the local maximum and local minimum points in the bearing stiffness curve disappear gradually (as shown in Figure 9 for a speed of 8000 rpm), and the shape of the bearing stiffness curve resembles that of the bearing stiffness curve at zero speed (similar to a "V" shape that slopes severely to the right). As the spindle rotates, the axial displacement at the minimum point of the spindle stiffness change rate curve is very close to the axial displacement at the point of abrupt change in spindle stiffness, so the spindle pre-compression displacement is generally much larger than the axial displacement corresponding to the minimum value of the spindle stiffness change rate curve. Thus, at the stage where the front and rear bearings of the spindle are jointly loaded, the increase rate of one-side bearing stiffness is always greater than the decrease rate of the other-side bearing stiffness, and the axial stiffness of the spindle increases with the gradual increase in the axial displacement. When the spindle axial displacement is greater than the pre-deflection of single-side bearing, the spindle bearing is completely unloaded, and the spindle axial stiffness is transformed into the other-side bearing stiffness, and the spindle axial stiffness increases with the gradual increase in axial displacement.

## 4. Experimental Verification

As indicated in Figure 10, a test rig was built to investigate the spindle axial operating stiffness experimentally. The fixed-position preload experimental spindle is a mechanical spindle that is driven by an electric spindle and has a variable-frequency motor to change the speed. During the test, the system control and data acquisition box's motor control switch can be rotated to change the direction of motor rotation on the axial loading device. Adjust the motor to rotate forward first, then apply axial pressure to the spindle with

the axial loading device. Adjust the motor reversal and the axial loading equipment to gradually release the axial pressure load to zero, and then provide a suitable amount of axial tension to the spindle. Finally, adjust the motor to rotate forward again, and the axial loading equipment will gradually reduce axial tension to zero. The force signal and related displacement signal are automatically captured in real-time by the NI®9215 data collection card during the test and sent to the computer for processing.

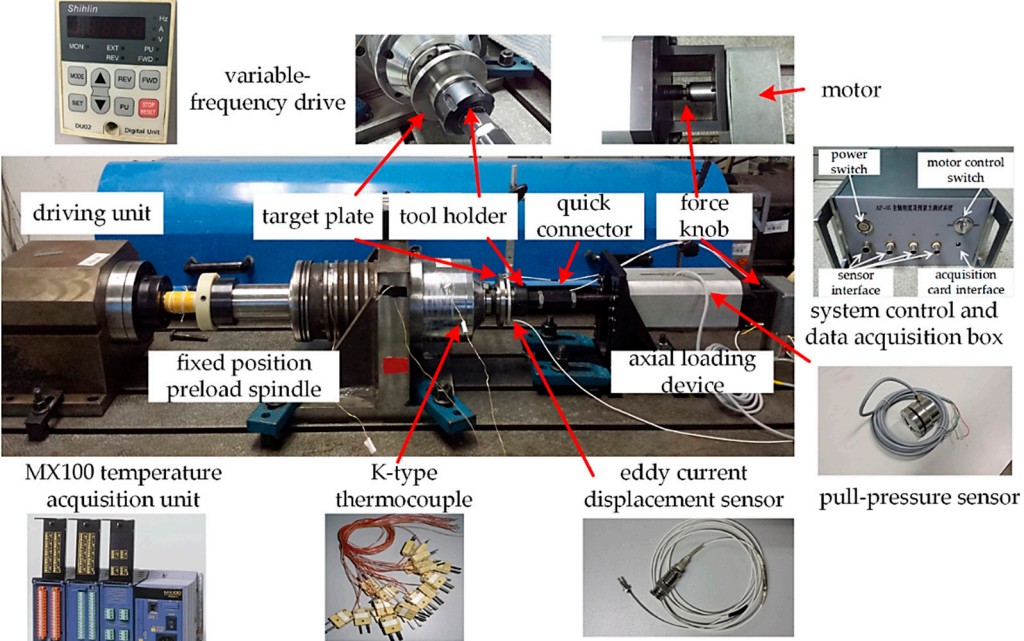

**Figure 10.** Spindle operating stiffness test rig.

The test spindle was set up in a back-to-back configuration, and the spindle's initial preload was set at 507 N by altering the width difference between the inner and outer spindle spacers. Before the test, the spindle components were cooled to room temperature. The influence of rotational speed on the spindle temperature field may be neglected here, since the test duration for each group is so brief. The spindle preload increases after the operation as a result of centrifugal load [29,30], and the change in spindle preload has a bigger impact on the spindle axial operating stiffness [4]. As a result, the real preload of the spindle at each speed of the experiment was computed during its execution. The test speeds were 0 rpm, 3000 rpm, and 6000 rpm, with the results displayed in Figure 11.

From Figure 11a–c, it can be seen that when the spindle axial displacement is small, the axial operating stiffness curve of the spindle with 0 rpm and 3000 rpm has certain fluctuations, but when the speed reaches 6000 rpm, the spindle axial operating stiffness curve fluctuation disappears, and the "sag" phenomenon appears, then the spindle exhibits a "stiffness hardening" characteristic. In addition, it can also be seen that the experimental results of the axial operating stiffness of the fixed-position preload spindle are consistent with the theoretical simulation results, and the two are in good agreement, where the maximum errors between the experimental and simulation results are 12.3%, 10.8%, and 10.5% for the spindle speeds of 0 rpm, 3000 rpm, and 6000 rpm, respectively. These errors can be attributed to three factors. To begin with, the filtered experimental data are distorted, and the filtering technique might be modified to lessen the error. Second, while bearing stiffness is the primary determinant of spindle stiffness, the experimental spindle stiffness also takes into account the stiffness of other spindle components, such as the draw bar mechanism. To lessen the error, the theoretical model can be enhanced further. Finally, there is a deviation between the real-time preload of the spindle and the true value of the preload in this experiment, which may be improved with more study into the transient preload measuring technique.

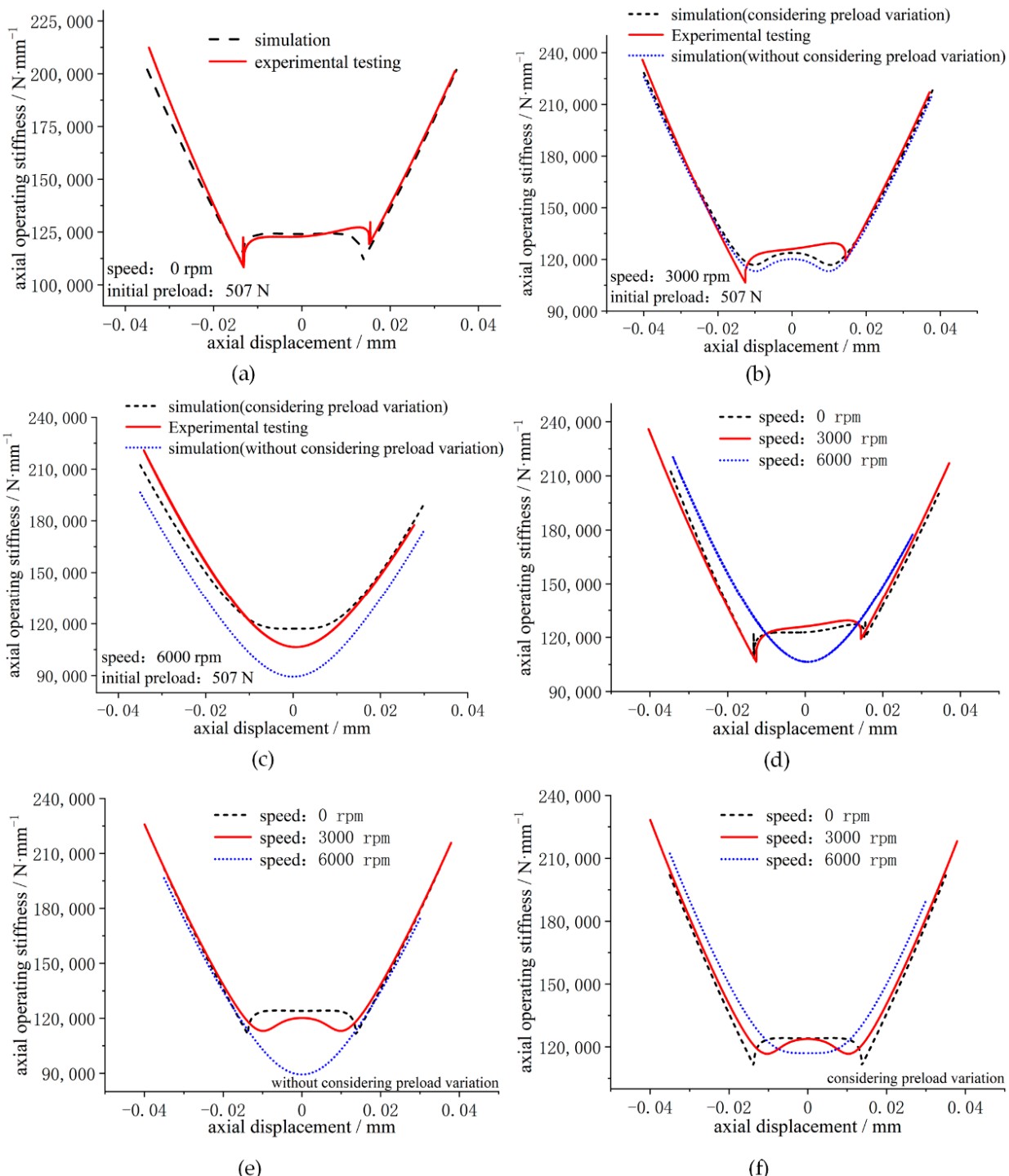

**Figure 11.** Experimental test on the operating stiffness of spindle at different speeds: (**a**) experimental testing (speed: 0 rpm); (**b**) experimental testing (speed: 3000 rpm); (**c**) experimental testing (speed: 6000 rpm); (**d**) experimental data comparison; (**e**) simulation data comparison (without considering preload variation); (**f**) simulation data comparison (considering preload variation).

As shown in Figure 11d, it can be seen that when the spindle speed is high, the spindle axial operating stiffness shows an obvious "sag" phenomenon, and the spindle exhibits the "stiffness hardening" characteristic. When comparing Figure 11d–f, it is clear that the experimental test findings in Figure 11d differ significantly from the simulation results, without considering preload variation in Figure 11e, but the trend is similar and less distinct from the simulation results when considering preload variation in Figure 11f,

indicating that the change in spindle preload due to the speed factor has a large impact on this experimental test of spindle axial operating stiffness and is an important factor causing the change in spindle axial operating stiffness.

## 5. Conclusions

In this paper, the axial operating stiffness model of the fixed-position preload spindle is proposed, and the effect of the rotational speed on the softening and hardening characteristics of the spindle stiffness is studied based on the model. An experimental bench for testing the axial operating stiffness of the spindle was built to verify the accuracy and effectiveness of the model. In the analysis of the experimental results, the influence of speed-induced preload on the spindle axial operating stiffness is specially analyzed. The conclusions of the article are as follows:

1.  For the fixed-position preload spindle with a smaller initial preload, the rotational speed has a greater influence on the load–displacement relationship and axial operating stiffness. However, the spindle speed has less effect on the load–displacement relationship and axial operating stiffness when the spindle's initial preload is greater.
2.  When the spindle is rotated, the axial displacement at the stiffness mutation point is greater than the axial displacement at zero speed. In the meantime, when the rotational speed increases, the axial displacement of the spindle's axial operating stiffness mutation point decreases.
3.  When the spindle is rotating slowly, the axial stiffness of the spindle fluctuates in the phase with small displacement increments. However, as the spindle speeds up, the stiffness fluctuation is gradually suppressed. When the spindle is operating at high speeds, the spindle axial stiffness displays a clear "sag" phenomenon, which shows the "stiffness hardening" characteristic.
4.  During the experimental test of the axial operating stiffness of the spindle, the change in spindle preload due to the speed factor has a large impact on test results and is also an important factor causing the change in spindle axial operating stiffness.

**Author Contributions:** Conceptualization, J.L., Q.W. and X.S.; Data curation, J.L., W.K. and S.M.; Formal analysis, J.L., J.Q. and A.Q.; Funding acquisition, J.L., W.K. and S.M.; Investigation, J.L., Q.W. and X.S.; Methodology, J.L. and J.Q.; Project administration, J.L., W.K. and S.M.; Resources, J.L., A.Q. and W.K.; Software, J.L.; Supervision, J.Q.; Validation, J.L., J.Q. and A.Q.; Visualization, J.L., J.Q. and A.Q.; Writing—original draft, J.L.; Writing—review and editing, J.L. and X.S. All authors have read and agreed to the published version of the manuscript.

**Funding:** This research received no external funding.

**Acknowledgments:** The authors would like to thank Yongsheng Zhu and Ke Yan from Xi'an Jiaotong University for their help in the experimental testing.

**Conflicts of Interest:** The authors declare no conflict of interest.

## Nomenclature

Capital Letter
$F_a$     Axial load applied to spindle
$F_{ab}$    Axial load applied to bearing
$F_c$     Centrifugal force
$F_p$     Preload applied to bearing
$Z$      Number of balls
Lowercase Letters
$a$      Distance between the raceway groove curvature center and the ball center
$d_m$    Diameter of bearing pitch circle
$m_b$    Ball mass
$n$      Spindle bearing speed

Greek Letters

$\alpha_f$      Initial contact angle of bearing

$\alpha_p$      Bearing contact angle with preload applied

$\alpha_i$      The ball-inner contact angle

$\alpha_o$      The ball-outer contact angle

$\delta$      Ball-race deformation

$\delta_a$      Spindle axial displacement

$\delta_{ab}$      Bearing axial displacement

$\delta_1$      Pre-deflection of front bearing

$\delta_2$      Pre-deflection of rear bearing

$\omega_c$      Bearing cage speed

Subscripts

$o$      Outer ring

$i$      Inner ring

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
