# Peer review of "Research on the Effect of Spindle Speed on the Softening and Hardening Characteristics of the Axial Operating Stiffness of Machine Tool Spindle"

_lubricants, doi:10.3390/lubricants10070132_

Round 1

Reviewer 1 Report

The current work invetigates the spindle axial operating stiffness of a fixed position preload spindle for precision machine tool. The axial stiffness is nonlinear and the effect of the rotational speed on the softening and hardening features of the spindle axial stiffness is studied. A numerical model is presented and its predictions are compared with measured results, with a good agreement. It is clear from the comparison of theoretical and experimental results that the preload variation due to the centrifugal loads affects the axial stiffness and must be considered in the modelling. Overall, the paper is well written and well organized. The quality of the paper can be improved if some points are elucidated:

1. Please, include a table of variables/nomenclature in the manuscript.

2. Is it possible to to include the the A0 and delta_ab variables of Eq. 1 in Fig. 1, to represent these variables in the bearing diagram?

3. In Table 1, is the inner and outer raceway radius of curvature equal? How can the inner and outer raceway radii be equal?

4. Following the discussion after Fig. 6, the authors mention the stiffness mutation point. It is not clear for me from Figs. 5-6 what the stiffness mutation point is. Please define and explain what is meant by 'stiffness mutation point'.

5. Throughout the manuscript, please check three things:

5.1. When refering to equations, figures or references, please include a space between the label and the number. In page 3, line 118, it is written 'from Eq.(3)' without a space. It should be 'Eq. (3)'. This also occurs in page 4, line 122 'Eq.(2)', line 124 'Eq.(4)', line 128 'Ref.[17]', line 135 'Eq.(5)', page 6, line 173 'Eq(12)' and so on. Please, revise it in the entire manuscript.

5.2. Do not separate in a line break the number and its units of measurment. use a hard space to keep the number and the units together. This occurs, for instance, in the legend of Figure 5 ('preload: 200 N)'. The number '200' is written in line 200 and its units of measurment 'N' is written on line 201. This also occurs in the legend of Figure 6 ('preload: 1000 N'). Please, revise it in the entire manuscript.

5.3. Sometimes throughout the manuscript the authors refer to Fig. X, while sometimes they write Figure X. Please, standardize it, either type the entire word or the abbreviation.

6. Please, increase the font size of the details shown in Fig. 7a and 9a.

7. A few minor typing errors were identified:

- Page 1, line 38, "Static and dynamic analysis are two approaches to studying spindle static stiffness." should be "[...] to study spindle static stiffness."

- Page 6, line 188, "When seen in Figure 5, [...]" should be "As seen in Fig. 5".

- Page 6, line 190, "the spindle speed has less of an effect on [...]" should be "the spindle speed has less effect on [...]"

- Page 12, line 330, "the spindle speed has less of an effect on [...]" should be "the spindle speed has less effect on [...]".

Please, check the entire manuscript for other typographical errors.

Reviewer 2 Report

Good approach about angular ball bearings.

No objections to the main contents and discussion, the only aspects to improve:

·         Spindles suffers under heavy operations and monitoring is the key to safe them in those unstable conditions.

·         One side of the spindle is completely unloaded during the pressure phase: (a) Fixed posi- tion preload spindle construction; (b)Spindle load distribution….Ok, this is classic theory, but: Please define if rotation is free in the working state. Is there any pivoting rotational speed component?

·         Some of the problems were defined by Recording of real cutting forces along the milling of complex parts, Mechatronics 16 (1), 21-32 because affected the surface of piece. Please extend the global view of machine tool applications.

·         Figure 3 and 4 are very good.

·         Values of ball bearing deformations affected several works, in new machine platforms. See Analysis of the tool tip radial stiffness of turn-milling centres, The International Journal of Advanced Manufacturing Technology 60 (9), 883-891

Conclusions are OK

Check several new sources: monitoring spindles were clasic in the 90s, Peña, Rivero, D Olvera, Urbikain,  S. Saravanan and others

Reviewer 3 Report

The authors investigated the influence of spindle speed on axial operating stiffness of the spindle by developing a mechanical model and the model was verified by experimental tests. I think, the subject is well described and the gap that this study will fill in the literature and the novelty of the study are well stressed. The findings are well presented and the reasons behind the phenomena are also well explained. Hovewer, some minör problems have to be solved before publication.

1. Literature review section should be expanded. I think, the number references is insufficient for such a good quality paper.

2. Although the mechanical modeling part is well explained and analyzed, the experimental part has not been disscused enough. This section must be detailed.

3. The authors stated “The experimental results of the axial operating stiffness of the fixed position preload spindle are consistent with the theoretical simulation results, and the two are in good agreement, where maximum errors between the experimental and simulation results 12.3%, 10.8% and 10.5% for the Spindle speeds of 0 rpm, 3000 rpm, and 6000 rpm, respectively.” But any discussion about the causes of these errors have not been made.

4. It would be better for the reader to provide a detailed explanation of what the "sag" phenomenon is.
